# Unmanaged Pharmacogenomic and Drug Interaction Risk Associations with Hospital Length of Stay among Medicare Advantage Members with COVID-19: A Retrospective Cohort Study

**DOI:** 10.3390/jpm11111192

**Published:** 2021-11-12

**Authors:** Kristine Ashcraft, Chad Moretz, Chantelle Schenning, Susan Rojahn, Kae Vines Tanudtanud, Gwyn Omar Magoncia, Justine Reyes, Bernardo Marquez, Yinglong Guo, Elif Tokar Erdemir, Taryn O. Hall

**Affiliations:** 1Invitae Corporation, San Francisco, CA 94103, USA; chad.moretz@invitae.com (C.M.); chantelle.schenning@invitae.com (C.S.); Susan.Rojahn@invitae.com (S.R.); 2OptumLabs at UnitedHealth Group, Minnetonka, MN 55343, USA; kaevinesgt@gmail.com (K.V.T.); GwynOmarMagoncia@uhg.com (G.O.M.); JustineReyes@uhg.com (J.R.); BernardoMarquez@uhg.com (B.M.); YinglongGuo@uhg.com (Y.G.); eliftokar@gmail.com (E.T.E.); tarynhall@uhg.com (T.O.H.)

**Keywords:** COVID-19, pharmacogenomics, medication management, hospitalization, precision medicine, length of stay, healthcare administration, healthcare costs, hierarchical conditions category (HCC), risk adjustment factor (RAF)

## Abstract

Unmanaged pharmacogenomic and drug interaction risk can lengthen hospitalization and may have influenced the severe health outcomes seen in some COVID-19 patients. To determine if unmanaged pharmacogenomic and drug interaction risks were associated with longer lengths of stay (LOS) among patients hospitalized with COVID-19, we retrospectively reviewed medical and pharmacy claims from 6025 Medicare Advantage members hospitalized with COVID-19. Patients with a moderate or high pharmacogenetic interaction probability (PIP), which indicates the likelihood that testing would identify one or more clinically actionable gene–drug or gene–drug–drug interactions, were hospitalized for 9% (CI: 4–15%; *p* < 0.001) and 16% longer (CI: 8–24%; *p* < 0.001), respectively, compared to those with low PIP. Risk adjustment factor (RAF) score, a commonly used measure of disease burden, was not associated with LOS. High PIP was significantly associated with 12–22% longer LOS compared to low PIP in patients with hypertension, hyperlipidemia, diabetes, or chronic obstructive pulmonary disease (COPD). A greater drug–drug interaction risk was associated with 10% longer LOS among patients with two or three chronic conditions. Thus, unmanaged pharmacogenomic risk was associated with longer LOS in these patients and managing this risk has the potential to reduce LOS in severely ill patients, especially those with chronic conditions.

## 1. Introduction

Hospitalizations represent nearly one-third of all U.S. healthcare expenditures [1]. Patient length of stay (LOS) is a longstanding measure of hospital efficiency as a longer LOS is associated with greater resource utilization and higher costs [2,3,4,5]. Among many factors associated with longer LOS are adverse drug reactions (ADRs), which are preventable with appropriate medication management [6,7,8]. Patient risk factors for ADR are inappropriate polypharmacy as well as the harboring of genetic variants that affect medication responses. Pharmacogenomics-guided medication management could therefore reduce both ADRs and LOS. Studies suggest that >99% of individuals harbor at least one variant associated with an atypical drug response and 24% of individuals may currently take a drug to which they may have an atypical response [9,10,11]. Given that a greater pharmacogenomic variant burden has been associated with longer LOS, [12,13] unmanaged pharmacogenomic risk may be an intervenable vulnerability for reducing LOS among high-risk patients.

Population ADR risk management could benefit hospital systems at any time and may benefit hospitals even more so when there is additional population-wide stress on the hospital system, such as the COVID-19 pandemic. The COVID-19 pandemic placed unprecedented demands for hospitals across the globe, and patient demand for hospitalization exceeded hospital resources in several regions [14,15,16,17,18]. In the US, roughly half of those hospitalized with COVID-19 have been Medicare and Medicare Advantage beneficiaries and the majority of these beneficiaries have been hospitalized for more than 5 days [19,20]. Medication management interventions may especially help the Medicare population as 54% of patients 65 and older report taking four or more prescription drugs daily [21]. This increases their risk for ADR-related hospitalization, which is reported to be four times greater among patients who are 65 years of age or older compared to younger patients [22]. Further, polypharmacy is a reported risk factor for developing COVID-19 [23]. Thus, hospitalized COVID-19 patients may be at heightened risk for ADR.

A recent study found that nearly 90% of patients hospitalized with COVID-19 had at least one medication order that could have been informed by pharmacogenomic testing and nearly one-quarter had orders for four or more actionable medications [24]. However, it is not clear whether pharmacogenomic risk or drug–drug interaction risk affects LOS among COVID-19 patients. To evaluate the impact of pharmacogenomic risk on LOS among high-risk patients, we conducted a retrospective analysis of administrative medical and pharmacy claims among Medicare Advantage members hospitalized with COVID-19. The main objective of this study was to determine if pharmacogenomic or drug–drug interaction (DDI) risk among Medicare beneficiaries hospitalized with COVID-19 was associated with LOS. We further examined this potential association by the number of diagnosed chronic conditions as well as four specific chronic conditions: chronic obstructive pulmonary disease (COPD), diabetes, hyperlipidemia, and hypertension.

## 2. Methods

### 2.1. Ethics Approval Statement

This study was granted a waiver of authorization by the institutional review board of the UnitedHealth Group Office of Human Research Affairs (FWA00028881, approved 23 February 2021) because the work is limited to a secondary analysis of de-identified claims data, and under the applicable exemption criteria, patient consent and HIPAA authorization are not required.

### 2.2. Data Source and Study Population

Data were obtained from an administrative database of medical and pharmacy claims from a large U.S. health insurance company. The claims include the International Classification of Diseases, Tenth Revision, Clinical Modification (ICD-10-CM) diagnosis codes, the National Drug Codes (NDC), patient demographic data, and other information.

Claims data for 10,206 Medicare Advantage members admitted to a hospital with COVID-19 between 1 January 2020 and 30 June 2020 were reviewed. COVID-19 diagnoses were identified through SARS-CoV-2 lab test records indicated by the Logical Observation Identifiers Names and Codes (LOINC) organization’s guidance for mapping SARS-CoV-2 and COVID-19-related LOINC terms [25]. Test information provided via LOINC terms indicated the test type (antibody, RT-PCR, etc.) as well as the test result (detected, not detected, not given/cancelled). Suspected COVID-19 inpatient cases were subjected to daily manual review by clinical staff via clinical notes to determine an individual’s COVID-19 status, flagging it as either negative, confirmed, presumed positive, or needing clinical review. All findings, except for confirmed status, were reviewed daily. Only patients’ first COVID-19 admission was included and evaluated, and patients who died during their hospital stay (*n* = 1604, or 16% of patients reviewed) were excluded from the study to prevent death from defining the outcome (LOS). Only patients continuously enrolled in Medicare Advantage Prescription Drug plans for at least 12 months prior to the index COVID-19 admission were eligible for inclusion. Pharmacogenomic testing was not an inclusion criterion for this study; instead, unmanaged pharmacogenetic risk was assessed as described below.

### 2.3. Independent and Outcome Variables

The primary independent variables were pharmacogenetic interaction probability (PIP) and DDI severity, as determined by Invitae’s YouScript clinical decision support tool. PIP is the probability that pharmacogenomic testing of 14 genes (*CYP2C19*, *CYP2C9*, *CYP2D6*, *CYP3A4*, *CYP3A5*, *CYP2B6*, *CYP4F2*, *DPYD*, *HLA-B*57:01*, *IFNL3*, *SLCO1B1*, *TPMT*, *UGT1A1*, and *VKORC1*) will result in the detection of one or more clinically significant pharmacogenomic interactions involving one or more medications on the patient’s current medication list. PIP was calculated based on publicly available phenotype prevalence data in the U.S. population and pharmacogenomic interaction data, which encompasses clinically actionable drug–drug or drug–drug–gene interactions in CPIC Guidelines and FDA labeling. More robust descriptions of YouScript are available in US patents [26,27]. PIP was categorized into the following: (1) low (PIP ≤ 25%), (2) moderate, (PIP 26–50%), and (3) high (PIP > 50%).

DDI indicates the potential for adverse interactions among 2 or more medications prescribed to a patient, irrespective of the patient’s pharmacogenomic profile. DDI severity was measured on an ordinal scale of increasing severity: minimal, minor, moderate, major, and contraindicated, with moderate or higher being tied to evidence indicating a need to consider changing a drug or dose. Among patients with multiple DDIs in their drug regimen, only the most severe DDI was used in this study.

All medications active or prescribed within 30 days prior to and during the first COVID-19 admission were screened for PIP and DDIs. Other independent variables were age, gender, race/ethnicity, residential location, median income, risk adjustment factor (RAF), hierarchical condition category (HCC), and Special Needs Plan (SNP) type. RAF scores and HCC, which are measures of disease burden used to determine the likelihood of a patient’s need for medical care, were based on the one-year window prior to the patient’s admission month. HCC are simplified diagnosis categories that are constructed using a standardized ICD-10 code mapping determined by the Centers for Medicare and Medicaid Services (CMS). HCC counts indicate the number of chronic conditions and were categorized as 0 or 1, 2 or 3, 4 or 5, or 6 or more, following previously reported CMS categorizations [28]. HCC, along with demographic data, were used in an algorithm to assign an RAF score, which is a measure of patient complexity [29]. SNPs are Medicare plan types restricted to individuals with specific diseases or characteristics. SNP types were: (1) chronic conditions SNP (C-SNP), for individuals with severe or disabling chronic conditions; (2) dual-eligible SNP (D-SNP), for individuals entitled to both Medicare and state Medicaid; and (3) institutional SNP (I-SNP), for individuals who need or are expected to need services in a long-term care skilled nursing facility or a similar high-level care facility for 90 days or longer [30]. The outcome variable of interest was LOS (in days).

### 2.4. Statistical Analysis

Zero-truncated negative binomial analyses (ZTNB) were used to assess the effects of independent variables on LOS. Two sets of ZTNB analyses were carried out. The first focused on the entire cohort and included the following models: (i) the baseline model that accounted for age, gender, PIP and DDI; and (ii) the regression model that adjusted for the variables identified by least absolute shrinkage and selection operator (LASSO) as covariates with a strong association with LOS [31]. The second set of analyses were subgroup analyses to limit the effects of underlying disease through model fitting and variable selection for each of the following: (1) patients stratified by HCC counts (i.e., number of chronic conditions); (2) patients stratified by each of the following chronic conditions regardless of other comorbidities: (i) COPD, (ii) diabetes, (iii) hyperlipidemia, and (iv) hypertension.

In both sets of ZTNB analyses, the point and confidence-interval estimate of the rate ratios and the corresponding *p*-values resulting from the fitted models were obtained. No medications were included as control variables in the regression models. Statistical significance was set at *p* < 0.05. Additionally, descriptive statistics were applied to profile the study population, including count, percentage, mean, and standard deviation (SD). Patients with any missing data, except for race/ethnicity, were excluded from the study cohort. Missing data on race/ethnicity are indicated as “Unknown” in Table 1. All analyses were conducted using the statistical computing software R, version 3.6.1 (R Foundation for Statistical Computing, Vienna, Austria) [32].

## 3. Results

A cohort of 6025 individuals met all inclusion criteria, including hospitalization with COVID-19 and 12-month continuous Medicare Advantage coverage (Figure 1). Most patients were female (61%), had a mean age of 77 years (SD, 11 years), were of white (non-Hispanic) ancestry (62%), lived in urban areas (47%), and had an average median household income of USD63,027 (SD, USD17,435) (Table 1). One-third of patients were enrolled in an I-SNP (34%), and 76% had two or more chronic conditions. More than half of the study population had hyperlipidemia (58%) or diabetes (52%).

Overall, 42% (2511 of 6025) of patients had a moderate or high PIP (Table 1), indicating that among these patients, there was a >25% chance that pharmacogenomic testing would reveal a clinically significant gene–drug or gene–drug–drug interaction. Further, 48% (2915 of 6025) had a moderate, major, or contraindicated DDI. Nearly one-quarter (24%) of patients (1422/6025) were prescribed metoprolol, a beta blocker associated with a minimum PIP of 46% (Appendix A). Other frequently prescribed drugs associated with moderate or high PIP were proton-pump inhibitors pantoprazole (12%, or 730/6025 patients) and omeprazole (9%, or 571/6025 patients), and serotonin-reuptake inhibitors escitalopram (6%, or 365/6025 patients) and citalopram (3%, or 207/6025; Appendix A).

Patients were hospitalized for a mean of 13 days (SD, 11 days; Table 1). Patients with moderate or high PIP were hospitalized for at least one day longer on average compared to those with low PIP, while LOS for patients with moderate, major, or contraindication DDI was similar to those with minimal or minor DDI (Figure 2 and Appendix A). In unadjusted analyses, the LOS for patients with moderate and high PIP were 9% (CI: 4–15%; *p* < 0.001) and 16% longer (CI: 8–24%; *p* < 0.001), respectively, compared to those with low PIP (Appendix A). Moreover, with other variables held constant, the LOS for male patients was 8% longer (CI: 3–13%; *p* = 0.001) compared to female patients, and for each one-year age increment, the expected LOS was 0.5% longer (CI: 0.3–0.7%; *p* < 0.001).

Among the entire cohort, adjusting for potential confounders did not attenuate the association between PIP and LOS (Table 2). When patients were stratified by HCC count (Table 2 and Appendix A), PIP was significantly associated with LOS among all patient groups except those with two or three conditions (Table 2). In particular, among patients with zero or one, or four or five conditions, those with moderate PIP had 15% (CI: 4–28%; *p* = 0.007) and 13% (CI: 2–25%; *p* = 0.019) longer LOS, respectively, compared to those with low PIP. Moreover, among patients with zero or one, or six or more conditions, those with high PIP had 39% (CI: 15–67%; *p* < 0.001) and 16% (CI: 3–31%; *p* = 0.014) longer LOS, respectively, compared to those with low PIP. Notably, across all patients, moderate and high PIP scores were significant predictors of LOS but RAF scores were not (Appendix A).

Varying associations were observed for LOS, HCC count, and DDI. Among patients with zero or one HCC, those with moderate or greater DDI had a 9% shorter LOS (CI: 0−18%; *p* = 0.045) compared to those with minimal or minor DDI (Table 2). Among patients with two or three HCC, those with moderate or more severe DDI had 10% longer LOS (CI: 2–18%; *p* = 0.010) compared to those with minimal or minor DDI. Lastly, across all HCC categories, enrollment in an I-SNP was associated with a 23–34% shorter LOS compared to those not enrolled in I-SNP (Appendix A).

To explore the impact of individual comorbidities on LOS, patients were stratified by four pre-existing chronic conditions: COPD, diabetes, hyperlipidemia, and hypertension. Patients with high PIP most frequently had hyperlipidemia (Appendix A). Within each chronic condition subpopulation (controlled for the other chronic conditions), patients with high PIP were hospitalized 12–22% longer compared to those with low PIP (Table 2). In particular, among patients with hypertension, those with high PIP were hospitalized 2.5 days longer than those with low PIP (Table 3). Moreover, patients diagnosed with hyperlipidemia or COPD who had moderate PIP had longer LOS by 8% (CI: 2–15%; *p* = 0.013) and 13% (CI: 3–24%; *p* = 0.009), respectively, compared to those with low PIP (Table 2).

When considering the potential impact of DDI on LOS, we found that patients diagnosed with diabetes and at least moderate or more severe DDI had 7% longer LOS (CI: 1–13%; *p* = 0.031) compared to those with minimal or minor DDI (Table 2). Lastly, patients in each chronic condition group who were enrolled in I-SNP had a 25–29% shorter LOS than those not enrolled (Appendix A).

## 4. Discussion

In this study, we showed that greater pharmacogenomic risk was associated with longer LOS among Medicare Advantage members hospitalized with COVID-19. In particular, high PIP patients, who have a 50% or greater likelihood of harboring a genetics-driven atypical response to current medications, were hospitalized for almost two days longer than those with low PIP. Given the known associations between ADRs and LOS, [8] our findings suggest that incorporating pharmacogenomic testing into the clinical management of high-risk COVID-19 patients may shorten hospital LOS. Critically, to potentially reduce LOS, testing would have to occur prior to admission to allow for testing results to be incorporated into a patient’s medication management.

In a novel pandemic such as the COVID-19 crisis, the sudden and intense demands on hospitals present a clear need to reduce patient LOS. Should proactive pharmacogenomics-informed medication management for COVID-19 patients successfully reduce LOS, even non-COVID-19 patients could benefit from the reduced burden on overwhelmed healthcare systems that have delayed some patients’ care [17,33,34,35]. Moreover, proactive management of pharmacogenomic risk could benefit many different patient populations through reducing hospital LOS. A study of patients hospitalized for a variety of conditions (mostly cancer and cardiovascular and kidney conditions) found that a higher number of CPIC-defined actionable variants was associated with a longer LOS [13]. Another study of patients with major depressive disorder showed that patients who harbored genetic variants associated with the reduced metabolism of some psychotropic medications and other drugs had longer LOS in psychiatric hospitals than patients without those variants [12].

In addition to potential benefits to patient health, pharmacogenomic intervention could reduce healthcare costs. Estimating the potential cost savings from a reduction in LOS is challenging; however, recent data show that the average Medicare payment per COVID-19 hospitalization is USD23,587, [19] which is USD8688 higher than the average Medicare Advantage hospitalization (USD14,900) [36]. Considering the higher cost associated with complex COVID-19 patients, pharmacogenomic-informed medication management of Medicare beneficiaries could result in substantial savings for patients, hospitals, and the Medicare program.

Overall, we found that almost half of Medicare Advantage members hospitalized with COVID-19 could benefit from improved medication management, as 42% had a moderate or high PIP and 48% had a moderate, major, or contraindicated DDI for current medications [9,10,11]. Further, despite mandates to flag DDIs in electronic health records and pharmacy systems, we found that one-third of our cohort was prescribed medications for which the risks likely exceeded the benefits. Thus, comprehensive medication management is underutilized among Medicare members, leaving many at risk for ADRs. Incorporating pharmacy expertise into the clinical management of patients could reduce this risk, as pharmacist participation has been shown to reduce ADRs, LOS, and mortality [37].

Several findings in our subpopulation analyses warrant further study. When patients were stratified by number of chronic conditions, patients with zero or one HCC and high PIP had a greater LOS increase than those with six or more HCC (39% versus 16% increase), which may be due to uncontrolled confounders such as the severity of the COVID-19 infection. In addition, we did not observe a significant association between PIP and LOS among patients with two or three chronic conditions, despite finding a significant association for all other HCC counts. A larger sample size may reveal significant findings for those with two or three HCC and possibly identify additional confounders. In addition, the finding that patients with zero or one HCC and more severe DDI had shorter LOS compared to those with less severe DDI requires further study that examines the specific medications used by those patients. Separately, for each specific chronic condition (COPD, diabetes, hyperlipidemia, and hypertension), we found patients with high PIP had significantly longer LOS compared to those with low PIP. Further research may explore the impact of pharmacogenomic risk and ADRs among COVID-19 patients with these comorbidities. Finally, in both subpopulation analyses, I-SNP enrollment was associated with shorter LOS. This may be a result of potential ADRs receiving more attention in institutional settings, or conversely, an effect of more expedient discharge due to enhanced care coordination with institutional settings.

LOS is a long-standing concern for healthcare organization performance, patient outcomes, and expenditures [38,39,40]. Although this study focused on COVID-19, population-based pharmacogenomic testing that improves medication management could broadly reduce ADRs, which are reported to cost the USD528 billion per year, exceeding the costs associated with any major disease or the actual drugs prescribed [41]. A one-time test could aid medication optimization and potentially reduce healthcare costs throughout each patient’s life. In addition, future research should further explore the potential for PIP to predict LOS, as moderate and high PIP were both significant predictors of LOS, whereas RAF score, a metric used by CMS for benchmarking expenditures and allocating payments to healthcare providers, was not (Appendix A).

Limitations to this study include that although this study showed an association between PIP and increased LOS, it was not a study demonstrating that increased LOS among COVID-19 patients was a direct result of adverse drug events that could have been prevented by proactive pharmacogenomic testing. Additional work, including prospective studies with large, controlled cohorts, is needed to demonstrate the impact of pharmacogenomics-guided medication management on hospital LOS. In addition, the analyses did not account for some potential confounders, such as the severity of COVID-19, the medication regimen complexity (including the types and counts of prescribed drugs, medication dosage or frequency), patient kidney or hepatic function, or patient response to active medications. Other potential confounders include the difference in racial and ethnic composition of the study cohort compared to the broad U.S. population frequencies used in the development of the PIP score. A general limitation of observational studies using claims data is that the presence of a code does not guarantee that a patient has been diagnosed or has received or taken a prescribed medication. Finally, our findings may have been affected by a survivor bias arising from the exclusion of patients who died during the study period.

## 5. Conclusions

Among otherwise risk-matched patients with COVID-19, a moderate or high pharmacogenomic risk was associated with longer LOS. Addressing pharmacogenomic risk prior to hospitalization for COVID-19 or other serious diseases may reduce LOS, decrease healthcare costs, and improve risk predictions in Medicare members, especially those with COPD, diabetes, hyperlipidemia, or hypertension.

## Figures and Tables

**Figure 1 jpm-11-01192-f001:**
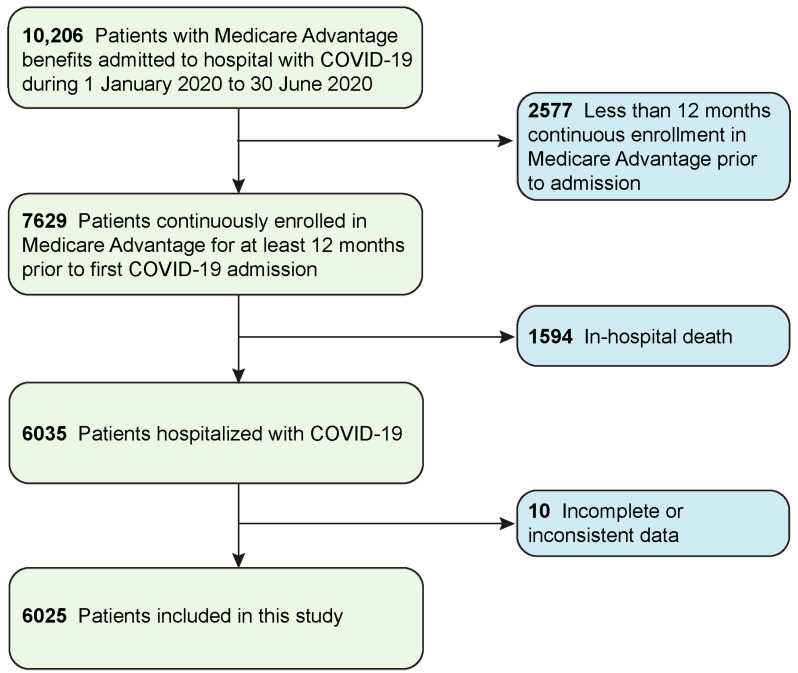
Selection of study patients.

**Figure 2 jpm-11-01192-f002:**
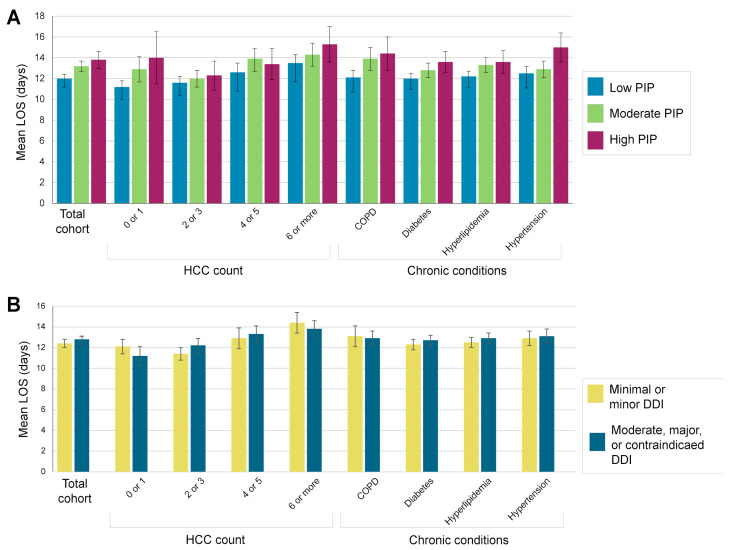
Length of stay by pharmacogenomic risk among Medicare Advantage members hospitalized with COVID-19. Average length of stay (LOS) in days among the entire cohort and subpopulations stratified by HCC count or chronic conditions was compared by (**A**) pharmacogenomic interaction probability (PIP) and (**B**) drug–drug interaction (DDI). Error bars show 95% confidence intervals. PIP: pharmacogenetic interaction probability; HCC: hierarchical condition categories; COPD: chronic obstructive pulmonary disease.

**Table 1 jpm-11-01192-t001:** Characteristics of hospitalized COVID-19 Medicare Advantage members.

Variable	Category	Value
Demographic/socioeconomic factors		
Gender	Female, No. (%)	3653 (61)
Age	Mean (SD)	77 (11)
Race/ethnicity	White (non-Hispanic), No. (%)	3753 (62)
	Black (non-Hispanic), No. (%)	1791 (30)
	Hispanic/Latino, No. (%)	191 (3)
	Other, No. (%)	140 (2)
	Asian/Pacific Islander, No. (%)	99 (2)
	Unknown, No. (%)	51 (1)
Residential location	Urban, No. (%)	2853 (47)
	Suburban, No. (%)	2200 (37)
	Rural, No. (%)	972 (16)
Median income	Mean (SD)	USD63,027 (USD17,435)
Plan and clinical characteristics		
C-SNP	Enrolled, No. (%)	268 (4)
D-SNP	Enrolled, No. (%)	332 (6)
I-SNP	Enrolled, No. (%)	2061 (34)
HCC count	0 or 1, No. (%)	1460 (24)
	2 or 3, No. (%)	1991 (33)
	4 or 5, No. (%)	1182 (20)
	6 or more, No. (%)	1392 (23)
Chronic conditions	COPD, No. (%)	1450 (24)
	Diabetes, No. (%)	3104 (52)
	Hyperlipidemia, No. (%)	3499 (58)
	Hypertension, No. (%)	1902 (32)
COVID-19 hospitalization (in days)	Mean LOS (SD)	12.6 (11)
PIP	Low, ≤25%, No. (%)	3514 (58)
	Moderate, 26–50%, No. (%)	1784 (30)
	High, >50%, No. (%)	727 (12)
DDI	Minimal or minor, No. (%)	3110 (52)
	Moderate, No. (%)	983 (16)
	Major, No. (%)	1542 (25)
	Contraindicated, No. (%)	390 (7)

C-SNP: chronic conditions special needs plan (SNP); D-SNP: dual-eligible SNP; I-SNP: institutional SNP; HCC: hierarchical conditions category; LOS: length of stay; PIP: pharmacogenetic interaction probability; DDI: drug–drug interaction.

**Table 2 jpm-11-01192-t002:** Expected length-of-stay ratios by pharmacogenetic risk and drug–drug interactions among patients by HCC count and chronic conditions based on regression models.

Population	Moderate PIP (26% to 50%) *		High PIP (>50%) *		Moderate, Major, or Contraindicated DDI **	
	Rate Ratio (95% CI)	*p*-Value	Rate Ratio (95% CI)	*p*-Value	Rate Ratio (95% CI)	*p*-Value
Total cohort	1.09 (1.04, 1.14)	<0.001	1.16 (1.09, 1.24)	<0.001	1.04 (1.00, 1.09)	0.066
HCC count						
0 or 1	1.15 (1.04, 1.28)	0.007	1.39 (1.15, 1.67)	<0.001	0.91 (0.82, 1.00)	0.045
2 or 3	1.03 (0.95, 1.11)	0.522	1.08 (0.96, 1.21)	0.204	1.10 (1.02, 1.18)	0.010
4 or 5	1.13 (1.02, 1.25)	0.019	1.13 (1.00, 1.29)	0.057	1.07 (0.97, 1.17)	0.179
6 or more	1.08 (0.98, 1.18)	0.112	1.16 (1.03, 1.31)	0.014	1.01 (0.93, 1.11)	0.786
Chronic conditions						
COPD	1.13 (1.03, 1.24)	0.009	1.18 (1.05, 1.34)	0.006	1.01 (0.93, 1.10)	0.796
Diabetes	1.05 (0.99, 1.12)	0.119	1.15 (1.05, 1.25)	0.002	1.07 (1.01, 1.13)	0.031
Hyperlipidemia	1.08 (1.02, 1.15)	0.013	1.12 (1.02, 1.22)	0.014	1.05 (0.99, 1.11)	0.096
Hypertension	1.03 (0.95, 1.12)	0.478	1.22 (1.10, 1.35)	<0.001	1.02 (0.95, 1.1)	0.522

* 0−25% as baseline; ** minimal or minor as baseline. HCCs: hierarchical conditions categories; PIP: pharmacogenetic interaction probability; DDI: drug–drug interaction; COPD: chronic obstructive pulmonary disease. Significance was set at *p* < 0.05.

**Table 3 jpm-11-01192-t003:** Distribution of length of stay among patients hospitalized with COVID-19 by chronic condition and pharmacogenomic or drug–drug interaction risk.

		Mean LOS (95% CI) by PIP			Mean LOS (95% CI) by DDI	
Chronic Condition	Total Subpopulation, Mean LOS (95% CI)	Low (≤25%)	Moderate (26–50%)	High (>50%)	Minimal or Minor	Moderate, Major, or Contraindicated
COPD	13.0 (12.4, 13.6)	12.1 (11.4, 12.8)	13.9 (12.8, 15)	14.4 (12.8, 16)	13.1 (12.1, 14.1)	12.9 (12.2, 13.6)
Diabetes	12.5 (12.1, 12.9)	12 (11.5, 12.5)	12.8 (12.1, 13.5)	13.6 (12.6, 14.6)	12.3 (11.8, 12.8)	12.7 (12.2, 13.2)
Hyperlipidemia	12.7 (12.3, 13.1)	12.2 (11.7, 12.7)	13.3 (12.6, 14)	13.6 (12.5, 14.7)	12.5 (12, 13)	12.9 (12.4, 13.4)
Hypertension	13.0 (12.5, 13.5)	12.5 (11.8, 13.2)	12.9 (12.1, 13.7)	15 (13.6, 16.4)	12.9 (12.2, 13.6)	13.1 (12.4, 13.8)

Data is presented in days. LOS: length of stay; PIP: pharmacogenomic interaction probability; DDI: drug–drug interaction.

## Data Availability

The data analyzed in this study was obtained from UnitedHealth Group Clinical Discovery Portal. The data are proprietary and are not available for public use, but, under certain conditions, may be made available to editors and their approved auditors under a data-use agreement to confirm the findings of the current study. Further inquiries can be directed to Lauren Mihajlov of UnitedHealth Group.

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
