# Peer review of "Unmanaged Pharmacogenomic and Drug Interaction Risk Associations with Hospital Length of Stay among Medicare Advantage Members with COVID-19: A Retrospective Cohort Study"

_jpm, 2021, doi:10.3390/jpm11111192_

Round 1

Reviewer 1 Report

This is a novel and interesting proposal of an observational pharmacogenomic study in COVID-19. Authors report the influence of the tool pharmacogenetic interaction study in the length of hospital stay of patients with COVID-19. The manuscript is adequately presented, nevertheless, the following comments must be addressed. 

  • The PIP is assessed considering risk variants in 14 genes that do not precisely interact with drugs administered in patients with COVID-19. First, there is no specific drug for COVID-19 treatment, and the drugs for the treatment of the symptoms could widely vary. Thus, the main responsible of the association of PIP risk with hospital stay length could be the drugs used for the comorbidities treatment. Did you verify if patients with comorbidities were controlled with the current pharmacological treatment?
  • There is an important proportion (30%) of African descendants among the included sample. Was the instrument validated in different populations, considering the interethnic variability in the frequencies and the clinical implications of the pharmacogenomics biomarkers?
  • Is there a comparison study including hospitalized patients with any other disease?
  • Line 117, HCC needs to be defined.
  • Table S3, footnote, PIP instead PPI. 
  • Limitation Section: Please, highlight that future pharmacogenomics studies should be prospective. 

Author Response

Reviewer 1:

  • The PIP is assessed considering risk variants in 14 genes that do not precisely interact with drugs administered in patients with COVID-19. First, there is no specific drug for COVID-19 treatment, and the drugs for the treatment of the symptoms could widely vary. Thus, the main responsible of the association of PIP risk with hospital stay length could be the drugs used for the comorbidities treatment. Did you verify if patients with comorbidities were controlled with the current pharmacological treatment?

Author response: The reviewer makes a great point; however, verifying if the comorbidities were controlled with active pharmacological treatment would require access to each patient’s electronic health record, which was not available. We have added a statement regarding this to the limitations section of the manuscript and clarified that no medications were included as control variables in the regression models.

Added text, lines 154–155: “No medications were included as control variables in the regression models.”

Added text, lines 339–340: “In addition, the analyses did not account for some potential confounders such as...patient response to active medications.”

  • There is an important proportion (30%) of African descendants among the included sample. Was the instrument validated in different populations, considering the interethnic variability in the frequencies and the clinical implications of the pharmacogenomics biomarkers?

Author response: The PIP score is calculated based on typical US population frequencies. While it is possible that the pharmacogenomic risk observed in the study could have been affected by the proportion of different ancestries in our cohort (the proportion of patients identifying as Black was over twice that of the US broadly), we do not believe this significantly affected the pharmacogenomic risk as no significant difference in LOS was observed across patient race/ethnicities (See supplemental Table 8). However, we have noted the potential confounding factor in the Limitations section.

Added text, lines 340–342: “Other potential confounders include the difference in racial and ethnic composition of the study cohort which differed from the broader U.S. population frequencies used in the development of the PIP score.”

  • Is there a comparison study including hospitalized patients with any other disease?

Author Response:  Yes, there is a recent study by Finkelstein et al that examined the impact of pharmacogenomic risk and LOS across a variety of patient types (PMID: 32604634) that was previously cited and another study by Ruaño et al that looked specifically at patient hospitalized with major depressive disorder (PMID: 23734807). We have added statements on both to the Discussion to make this more clear and have cited both in the introduction (line 40). Please note this required updates to the numbering of many references.

Added text, lines 270–277: “Moreover, proactive management of pharmacogenomic risk could benefit many different patient populations through reduced hospital LOS. A study of patients hospitalized for a variety of conditions (mostly cancer and cardiovascular and kidney conditions) found that a higher number of CPIC-defined actionable variants was associated with longer LOS.(Finkelstein et al. 2020) Another study of patients with major depressive disorder showed patients who harbored genetic variants associated with reduced metabolism of some psychotropic medications and other drugs had longer LOS in psychiatric hospitals than patients without those variants.(Ruaño et al. 2013)”

  • Line 117, HCC needs to be defined.

Author Response: Thank you. We added the definition at that first mention (line 126).

  • Table S3, footnote, PIP instead PPI.

Author Response: Correction made.

  • Limitation Section: Please, highlight that future pharmacogenomics studies should be prospective.

Author Response:  We clarified that future research should be prospective in the Limitations section.

Added text, line 334: “Additional work including prospective studies with large, controlled cohorts is needed to demonstrate the impact of pharmacogenomics-guided medication management on hospital LOS.”

Reviewer 2 Report

In this manuscript the authors underlighted the association between the unmanaged pharmacogenomic and drug interaction risks determined by Invitae’s YouScript clinical decision support tool and the lenghts of stay among patients hospitalized with COVID-19. The results highlighted that greater pharmacogenomic risk in particular in high PIP patients could prolonged LOS of almost two days compared to low PIP patients. It is well written and provides very interesting information regarding the importance of pharmacogenomic analyses also in the management of hospitalized patients with COVID-19.

I have a few of minor comments:

1) In Table S6, S7, S8, S9, S10 the type of model used could be specified in the title.

2) The list of HCC considered in Table S6 not reported all the HCCs reported in Table S7, S8, S9 and S10 (for example, the HCC010 in Table S7 lack in Table S6). I suggested to check if the data reported in these tables are correct.

3) The information reported in Table 3 and Table S5 could be merged in a single table, similarly to Table 2.

4) At line 168-171: add to the percentages the number of patients.

5) The reference No. 1 should be updated with “Overview of U.S. Hospital Stays in 2016: Variation by Geographic Region” (https://www.hcup-us.ahrq.gov/reports/statbriefs/sb246-Geographic-Variation-Hospital-Stays.jsp).

Author Response

1) In Table S6, S7, S8, S9, S10 the type of model used could be specified in the title.

Author Response: Thank you. We have added the type of model (Zero-truncated negative binomial (ZTNB) regression model) to the titles of the supplemental tables requested as well as Tables 2–3, S3, and S11–S14 (formerly S12–S15).

2) The list of HCC considered in Table S6 not reported all the HCCs reported in Table S7, S8, S9 and S10 (for example, the HCC010 in Table S7 lack in Table S6). I suggested to check if the data reported in these tables are correct.
Author Response: Due to differences in variable selection made by LASSO in the different subgroups of the cohort, it is possible for a variable such as HCC to be relevant to one analysis and not another and thus presented in one table and not to be reported in another table. The subgroup used in Table S6 (now Table S5) consists of all patients regardless of HCC count, whereas the subgroup used in Table S7 (now Table S6) consists of all patients with zero or one HCC. We have added a footnote to the tables to highlight this for readers.

Added text (Table S5–S9 footnotes): “Variables shown were determined by least absolute shrinkage and selection operator analysis, which can vary by cohort subgroup.”

3) The information reported in Table 3 and Table S5 could be merged in a single table, similarly to Table 2.
Author Response: Thank you for the suggestion, we have combined the tables. Please note this resulted in a change to the numbering for many supplemental tables.

4) At line 168-171: add to the percentages the number of patients.

Author Response: The numbers have been added to the text (lines 187–191).

5) The reference No. 1 should be updated with “Overview of U.S. Hospital Stays in 2016: Variation by Geographic Region” (https://www.hcup-us.ahrq.gov/reports/statbriefs/sb246-Geographic-Variation-Hospital-Stays.jsp).

Author Response: We could not find data in the suggested link to support the statement that hospitalizations represent nearly one-third of all U.S. healthcare expenditures. However, we did find a more recent citation than the one included in our original submission and have updated the reference. (“Concentration of Healthcare Expenditures and Selected Characteristics of High Spenders, U.S. Civilian Noninstitutionalized Population, 2018,” PMID: 31083861)

Round 2

Reviewer 1 Report

Thank you for the modifications performed to the manuscript.